# Living Conditions and the Incidence and Risk of Falls in Community-Dwelling Older Adults: A Multifactorial Study

**DOI:** 10.3390/ijerph20064921

**Published:** 2023-03-10

**Authors:** Irene Escosura Alegre, Eduardo José Fernández Rodríguez, Celia Sánchez Gómez, Alberto García Martín, María Isabel Rihuete Galve

**Affiliations:** 1Nursing and Physiotherapy Department, University of Salamanca, 37007 Salamanca, Spain; 2Institute of Biomedical Research of Salamanca, 37007 Salamanca, Spain; 3Department of Developmental and Educational Psychology, University of Salamanca, 37007 Salamanca, Spain; 4Department of Labour Law and Social Work, University of Salamanca, 37007 Salamanca, Spain; 5Medical Oncology Service, Salamanca University Hospital, 37007 Salamanca, Spain

**Keywords:** older adult, active aging, scale, risk of falls, life conditions, nursing role

## Abstract

Old age represents a social group that is undergoing continuous expansion. The aging population will be prone to chronic diseases and falls, which is a marker of frailty and a public health problem. This study aims to examine the relationship between living conditions and the prevalence of the risk of falls in older adults within the community. As an observational cross-sectional study, intentional sampling was carried out on residents of the metropolitan area over 75 years of age. The socio-demographic data of the subjects and their history of falls were collected. Additionally, the subjects were evaluated on the risk of falling, basic activities of daily living, such as walking and balance, fragility, and their fear of falling. The statistical analyses used were based on the Shapiro–Wilk test for normality, statistics of central tendency with description, mean (M) and dispersion, standard deviation (SD), bivariate contingency tables for studying the relationships between the variables, and the analysis of Pearson’s relational statistics (χ^2^). The comparisons of means were resolved by parametric or non-parametric routes. We obtained the following results: 1. The socio-demographic profile of our sample consisted of adults over 75 years of age, the majority of whom were overweight or obese women living in an urban area, specifically in an apartment, and receiving care; 2. Older people in the studied community had mild dependency and frailty, and were also at severe risk of falls; 3. The prevalence of falls was higher in women than in men in this study. Through these results, we confirmed the relationship between living conditions and the prevalence of risk of falls in older adults within the community.

## 1. Introduction

Different authors [1,2] understand old age as a demographic force, a social group in continuous expansion. This situation will continue in the upcoming years as there is a low birth and mortality rate. The constant increase in life expectancy produces an inversion in the population pyramid, and this shows a clear process of demographic revolution, which has been called the “Silent Revolution” [3]. The increase in the population group aged over 80 years has been identified as the Aging of the Elderly [4]. It has been estimated that by 2030, one in six people in the world will be 60 years of age or older [5]. It is also expected that by 2050, the world population of people in this age group will double (2.1 billion), and the number of people aged 80 and over will triple between 2020 and 2050, reaching 426 million [5].

There is a common phenomenon that is affecting a large proportion of older adults: fall incidence. Falls among the elderly are an important health problem due to their high prevalence and serious consequences: immediate physical consequences (e.g., fractures, bruises, and chest trauma), late physical consequences (e.g., prolonged stay on the floor, which causes dehydration, infections, and immobility), psychological consequences (e.g., post-fall syndrome), and social and economic consequences, which can also entail a high mortality rate [6]. Falls are associated with restricted mobility, decreased ability to perform daily activities, loss of security, fear of falling again, depression, and an increased risk of dependency [7]. Falls have an impact on the entire population, but older adults are the ones at the highest risk of fatal falls [8]. Falls are the leading cause of accidental death in older adults, being responsible for 70% of accidental deaths in people over 75 years of age [9]. 

Falls among the elderly represent a public health problem worldwide due to their frequency, the associated morbidity and mortality, and the high cost of health resources that they entail [10,11]. Falls incidence is usually the result of a complex interaction between intrinsic factors, extrinsic factors, and circumstantial factors, and there are even other types of factors that have given rise to what some authors have called “unclassifiable falls” [12]. 

The World Health Organization (WHO) defines a fall as “the consequence of any event that plunges the patient to the ground against his will.” This event is usually sudden, involuntary, and unsuspected, and can be confirmed by the patient or by a witness [13]. Falls make up one of the so-called Great Geriatric Syndromes, or Geriatric Giants, and they themselves represent a marker of frailty [14]. Falls have an impact on the entire population, but older adults are the ones at the highest risk of fatal falls [8].

The main objective of this study was to know, in detail and comprehensively, the relationship between living conditions and the prevalence of risk of falls in older adults within the community of Salamanca.

## 2. Materials and Methods

The study design is observational and cross-sectional and is carried out on the population of Salamanca within their Metropolitan district and Public Health System. The selection of the samples was made by the residents of the metropolitan area of the community of Salamanca (refer to Table 1).

The sampling technique employed in this study was intentional. Those who met the inclusion criteria and who did not present any of the exclusion criteria were recruited.

The calculation of the sample size was conducted through the formula for the estimation of a proportion for infinite populations. The value of n (377.67 people) has been rounded up, identifying 378 people as the minimum size. The number of participants was 425 (n = 425); however, there were 10 excluded (n = 10), so the total of participants was: 415 (n = 415). 

The variables to be studied are fear of falling, risk of falling, gait and balance, performance of basic activities of daily living (autonomy), and physical performance. To assess the variables, we used the following average instruments: -*Fear of Falling Test:* instrument to assess the prevalence of fear of falling. [15,16,17]. To measure this phenomenon, the question “Are you afraid of falling?” with ordinal response: nothing, a little, moderate, and a lot of fear. Additionally, we used the question “Have you limited your activities for fear of falling?” with yes/no answer. Additionally, the Short FES-I scale was also carried out, with seven items, whose score ranges from 7 to 28 points, the higher the score, the greater the concern related to falls.-*Tinetti Test:* instrument to assess gait and balance [18]. The maximum score that can be obtained is 28 points. There is a high risk of falling when the score is <19, moderate risk: 19–24 points and low risk: >24 points.-*SPPB Test*: instrument to assess the frailty of the person. (It was not used in any calculation due to the inability to differentiate the people studied; all were shown as frail people) [19]. The total score ranges from 0 to 12. Depending on the result obtained in the test, the elderly can be classified as a non-fragile autonomous person if the score is ≥10 points, and a frail person with <10 points.-*Barthel Test*: instrument for the assessment of the basic activities of daily life [20]. The results are grouped into dependency categories: total (<20), severe (20–35), moderate (40–55), mild (≥60), and independence (100) [20].-*Downton Test:* instrument to assess the risk of falling [21,22]. The scale has five sections, and each one is scored with 1 if the referred condition is present and 0 if not. The total score is from 0 to 11. If the total result is ≥3, it indicates a high risk of falls.

In the information gathering process, first, the work material was prepared: Data collection sheets (source of information) with the assessments made throughout the interview in paper format. The socio-demographic data of the subjects, pathological history, history of falls, and habitual medication were collected. The results obtained were gathered, which were processed through a statistical analysis to later interpret them.

For the ethical–legal aspects, confidentiality of data and results, we took into account the rules of special protection that Organic Law 15/1999, of December 13, Protection of Personal Data (LOPD) establishes for health data, Royal Decree 1720/2007, of 21 December, which approves the Regulations for the development of Organic Law 15/1999, and the instructions given by the SAS General Secretariat (21 August 2007) to ensure that the provisions of the LOPD are complied with in its centers sanitary. The personal and affiliation data remained in the custody of the person in charge of the district appointed for this purpose. 

This study was approved by Clinical Trials with the registry number: 0000281.

Each subject included in the study was identified with a unique number (the selection number), to guarantee anonymity.

For the statistical methodology, first, the results obtained from the data collection and tests conducted were digitized to prepare a data matrix, and thus, be able to carry out a descriptive analysis of all the variables and data collected in the evaluation.

The next step was the classification of each variable in a double aspect (qualitative and quantitative), describing the former by means of frequencies and percentages, with bar chart graphs. As a step prior to the study of the quantitative variables, the assumption of normality was analyzed with the Shapiro–Wilk test; later, they were described with the statistics of central tendency, mean (M) and dispersion, and standard deviation (SD).

The relationships between variables were analyzed using the strategy of bivariate contingency tables (qualitative and/or ordinal variables) and the analysis of Pearson’s relational statistics (χ^2^). In the case of normal quantitative variables (parametric pathway), the relationships were analyzed with the Pearson (r) correlational strategy, and non-normal (non-parametric pathway) with the Spearman (ρ) correlational strategy.

The comparisons of means were resolved by the parametric or non-parametric route depending on compliance with the previous assumption of normality. For the parametric pathway and in the case of comparing two means, they were resolved with the Student (t) test, and in the case of comparing three or more means, they were resolved with the Snedecor (F) test. For the non-parametric pathway and in the case of the comparison of two means, they were resolved with the Mann–Whitney (U) test, and in the case of the comparison of three or more means, they were resolved with the Kruskal–Wallis (H) test.

In all cases, a type I error (α risk) of 0.05 (5%) has been considered, that is, the confidence interval is 95% with a significance index of *p* < 0.05.

The calculations were developed with the statistical package IBM-SPSS Statistics, version 19.

## 3. Results

The analysis of the socio-demographic characteristics and the descriptive values of the age, height, and BMI (Table 2), besides the chronic diseases and the medication (Table 3) of the studied population, whose computation is that collected through the previously established inclusion criteria, bringing together a total of 415 people, offers the following results: 

Fear of falling test:

When asked “Are you afraid of falling?” we obtain (Figure 1):

When asking the people participating in the study, “Does the fear of falling limit your activities?” (Figure 2).

Within the limitation of activities due to fear of falling, we find that 244 people (58.8%) have a moderate fear of falling from bathing or showering, 204 people (49.2%) have a moderate fear of falling from going up–down stairs, and 220 people (53%) have a moderate fear of falling for going out to an event. A total of 283 people (68.2%) have little fear of falling from dressing–undressing, 278 people (67%) have little fear of falling from sitting-getting up from a seat, and 246 people (59.3%) have little fear of falling from picking things up or on the ground.

The record of each patient is completed with the instrumentation of the five objective tests, tests, to which this work has already referred previously. All of them, after conducting normality tests, are not normal.

To the question “Have you had falls?”, 226 people (54.5%) answered no and 188 (45.3%) answered yes, and as shown in Figure 3, the recorded number of falls has been: 

When analyzing the relationships that could occur between having had falls and the gender variable (Table 4), it is confirmed that there is a relationship between the two (χ^2^ = 20.86, *p* < 0.001). Thus, we see that of the 245 women 134 (54.69%) have fallen, while of the 169 men, only 54 (31.95%) fell.

There is also a relationship between falls and having had support (χ^2^ = 9.42, *p* < 0.001), that is, of the 226 people who have not fallen, 128 do not have support and 98 do; additionally, of the 188 people who did fall, 78 people do not have support and 110 do (Table 4).

Significant differences are found between height (t = 2.07 and *p* = 0.038) and age (U = 17,277.50 and *p* = 0.001), depending on whether they have fallen people who are shorter and older fall more (refer Table 5).

Studying the relationship between having had falls and chronic disorders (Table 6), it is possible to show an existing relationship between having had falls and disorders related to the nervous system (χ^2^ = 7.46, *p* < 0.001). Of people who did not suffer from this type of chronic disorder, 142 (62.8%) had not fallen and 84 (37.2%) had, while of those who did, 93 (49.5%) did not fall and 95 (50.5%) did.

In the same way, a relationship can be observed between having had falls and chronic psychological disorders (χ^2^ = 4.33, *p* < 0.001). Of the people who did not suffer from this type of psychological disorder, 123 (54.4%) did not had fallen and 103 (45.6%) had, while of those who did, 83 (44.1%) did not fall and 105 (55.9%) did.

Additionally, having had falls and disorders related to the female genital tract and the male genital tract are related. In the case of the female, (χ^2^ = 11.39, *p* < 0.001). Of the 226 women who did not suffer from this type of disorder, 202 (89.4%) had not fallen and 24 (10.6%) had, while of the 188 people who did suffer from it, 145 (77.1%) did not fall and 43 (22.9%) did fall. In the case of men (χ^2^ = 10.48, *p* < 0.001), of the 226 men who did not suffer from this type of disorder, 155 (68.6%) had not fallen and 71 (31.4%) had, while of the men 188 who did suffer from it, 155 (82.4%) did not fall and 33 (17.6%) did fall.

There is a relationship between having had falls and the consumption of medication (Table 7) for the nervous system (χ^2^ = 3.60, *p* = NS) and the respiratory system (χ^2^ = 7.095, *p* < 0.001).

Deepening into the relationship between the intensity of fear of falling and the occurrence of falls (Table 8), we show that a relationship can be established between both variables (χ^2^ = 197.63, *p* < 0.001). 

Quantifying the relationship between the limitation of activities of daily living due to fear of falling and the occurrence of falls, we show that a relationship can be established between these variables (χ^2^ = 56.38, *p* < 0.001). Of the 226 people who have not fallen, 200 individuals (88.5%) have not limited daily activities for fear of falling, and 26 individuals (11.5%) have. Of the 188 people who did have falls, 105 (55.9%) have not limited daily activities for fear of falling and 83 have (44.1%).

Observing the relationship between the fear of falling test and the occurrence of falls, we show that a relationship can be established between both variables (χ^2^ = 114.74, *p* < 0.001). Of the 226 people who have not fallen, 142 (62.8%) are not afraid of falling, while of the 188 people who have fallen, 167 (88.8%) are very afraid or moderately afraid of falling.

Studying the relationships between having falls and the tests (Table 9), it is verified that there is a relationship between the variables having had falls and the Tinetti Test (X^2^ = 13.32, *p* < 0.001). In the moderate risk group, there are more people who have not suffered falls compared to those who have, 114 (50.2%) compared to 61 (32.4%). In the case of the severe risk group, a certain difference can be seen between both groups, 113 people have not fallen (49.8%) and 127 (67.6%) have fallen.

Observing the relationship between the Barthel test and the occurrence of falls (X^2^ = 9.03, *p* = 0.029), it can be seen that there is a relationship between both variables; specifically, of the 226 people who have not fallen, 32 people (14.2%) are independent, 189 people (83.6%) are lightly dependent, and 5 people (2.2%) are moderately dependent. Of the 188 people who did fall: 19 people (10.1%) are independent, 154 people (81.9%) are lightly dependent, 13 people (6.9%) are moderately dependent, and 2 people (1.1%) are severely dependent. Independent people do not fall as much, compared to severely, moderately, or slightly dependent people. In other words, the greater the dependency, the greater the proportion of falls that occur.

Studying the possible relationship between the Downton-3 test (recoded at two levels) and the occurrence of falls, we show that a relationship can be established between both variables (X^2^ = 22.79, *p* < 0.001); more specifically, of the 226 people who have not fallen, 41 (18.1%) are not at risk of falls and 185 (81.9%) are at risk. Of the 188 people who did fall, 6 (3.2%) are not at risk of falls, but 182 (96.8%) are at risk of falls.

## 4. Discussion

The mean age of our study was 82.63 years old, according to the inclusion criteria. This coincides with the CSIC [23] report in 2020, in which they indicate that the proportion of octogenarians continues to increase to a greater extent in Spain. Most of the older people in the current study are women. This distribution by age and sex (in our study: n = 245, 59%) maintains the same trend as that presented in other studies. This phenomenon alludes to the term known as Feminization of old age [23]. Taking as reference Eurostat, Healthy life years, based on the Living Conditions Survey [23], overweight and obesity are two characteristics present in the elderly [23]. In our study, we have verified this reality (BMI, X = 27.82, SD = 4.49), which is why we consider that nutritional status is a crucial factor to take into account due to its impact on the health and functionality of the elderly. On the other hand, the height of our sample has an average of 1.57 m, providing a relevant and curious fact: there are significant differences between height and age depending on whether or not they have had falls, being quantitatively more significant in people with less height and with greater age, as we have already commented in the results section. Despite being considered for a more exhaustive analysis of the characteristics of falls, we have not found this point in the specialized literature. Considering the marital status of our subjects, it can be seen that most of them are married, followed by widows and to a lesser extent single. The explanation can be found in the fact that the majority of the population object of this study belonged to the male sex, as in Spain in 2018, where the percentage of married men exceeds that of women in all age groups of 65 and over, exceeding the rest of the marital status of men in the same way [23]. However, this situation differs from other studies [24,25] and may be due to the fact that the populations compared were not homogeneous. Regarding the type of residence and cohabitation, it can be verified that most of the population in our study lives in urban areas and in flats, contrasting with another study on the presence of frail elderly in the population, carried out in Guadalajara [26], in which half of the population studied lived in rural areas.

Chronic diseases, such as the presence of the pathology of Arterial Hypertension (HTA), are the majority in our sample (F = 391, 94.2%). This data coincides with the prevalence in the population over 60 years of age in Spain [27]. In this sense, there are studies that evaluate the association between the consumption of different antihypertensive drugs and the occurrence of falls, obtaining disparate results [28,29]. A relationship between the use of antihypertensives and the risk of serious injuries due to falls has not been proven [30]. Several lines of research suggest that antihypertensive drugs may increase both the risk of falls and injuries from falls in older adults. 

Most of the population in our study suffers from locomotor system problems (F = 389, % = 93.7). Coinciding with EESE 2020 [31], which indicates that they are among the most frequent diseases or chronic health problems suffered by the population: osteoarthritis, low back pain, and cervical pain. According to the World Health Organization (WHO), osteoporosis affects 3.5 million people in Spain. In 2010, it was estimated that, in the European Union, a total of 22 million women and 5.5 million men had been diagnosed with densitometric osteoporosis [32]. Among the fractures, the most common are hip and vertebral. In Spain, a prevalence of 104 cases per 100,000 inhabitants is estimated, which means 45,000 to 50,000 hip fractures per year with an annual cost of EUR 1591 million and a loss of 7218 quality-adjusted life years. 

Most of the participants present pathologies related to the endocrine system (F = 358, % = 86.3), coinciding with the national data, since more than a third of the population over 75 years of age suffer from diabetes [33]. One of the pathologies implicated in the risk of falls and that interferes with the autonomy of the basic activities of daily life is the sensory deficit and, specifically in our study, the visual deficit. 65.1% (F = 270) of the participating older adults suffered from it, a prevalence rate similar to that of other investigations [34,35], which is why it also constitutes one of the main aspects in terms of fall prevention. The fact that in our study, we did not find high rates of cognitive impairment, unlike other works [36], may be because those people who had severe cognitive impairment did not fall within the inclusion criteria of the research. However, the prevalence of psychological disorders in our study does not differ from other investigations [37,38]. The results provided in our study on genitourinary problems present in more than half of the population analyzed (F = 253, 61%) agree with other studies at the national level [39,40]. These conditions interfere with the quality of life of the elderly, also implying a significant social, economic, and health cost.

Medication consumed: the type of medication consumed can pose a greater risk in the production of falls, as is the case with psychoactive drugs [41,42]. It has been possible to verify that the following two types of medications, hypnotics and anxiolytics, and antidepressants, have been independently associated, by 50% each, with the probability of falls [43]. The use of digoxin, a type of IA antiarrhythmic and diuretic, has also been associated with an increased risk of falls [44]. and the use of laxatives [45]. The results obtained in our study coincide with the rest of the investigations, the most used have been those related to the cardiovascular system (F = 364, 87.7%), followed by those that treat pathologies derived from the nervous system (F = 343, 82.7%), and finally, those related to the alimentary tract and metabolism (F = 311, 74.9%). We can also mention that the elderly in our study mostly consumed more than four medications. It has been discovered that there is a consistent relationship between polypharmacy and falls in the elderly. There have been several studies [44,45,46] that have considered taking four or more medications as important predictors, although this may be due to the number of associated chronic processes [46].

Most of the studies that analyze the degree of autonomy of the patients do so using the interpretation suggested by Sah et al. [47]; however, we have decided to use the interpretation used by Carballo et al. [48] because the ranges between the scores are less limited than in Shah’s [44]. Based on our results, the majority are mildly dependent (82.9%). It is noteworthy how some of the people who receive a dependency score manage to hide this characteristic by developing alternative capacities for the execution of BADLs.

In the present study, we can observe with Tinetti Test that most adults in our environment present a severe risk of falls (57.8%), data similar to the study by Carballo-Rodríguez et al. [48], since 50% of the sample presented a high risk of falls.

It must be taken into account that there is a disparity in scores in three items compared to the original version of the Spanish translation in this scale, specifically in: “other medications”, “safe walking with help”, and “impossible walking”, since the author assigned them the value of 0 in the original version; however, in the version translated into Spanish, a value of 1 is provided if the conditions of each item are present. This fact was developed in a systematic review conducted by Aranda-Gallardo et al. [21]. In the present study, the Spanish translation has been used and the results have reflected how the majority, 368 people, present a risk of falls, compared to the 47 who do not. It has been possible to verify how this instrument is not recommended for use in the hospital setting, since the ability to predict the risk of falls decreases significantly. However, outside of this setting, there is no evidence to rule out its use. On the contrary, in a study [49] carried out with Swedish older adults, it has been shown how this test could be capable of independently predicting injuries related to falls, brain injuries, hip fractures, and mortality in older people.

In a systematic review [50], where the validity of nine instruments was evaluated, including the modified versions, to assess the physical conditions of people aged 60 or over who live in the community, it was possible to verify how this battery is highly recommended, both in terms of validity, such as reliability and responsiveness. Regarding the results obtained in this test (100%), only one level is detected: fragile people. Values shared in other previously conducted studies [19,51] are also in the community. It must be taken into account that the score we used is the same as in the study by Cabrero-García et al. [19], that is, categorical according to the execution intervals (0–4 points), unlike Abizanda Soler et al. [51] who use continuous scores based on the execution time of the tests.

The fear of falling again is the main psychological consequence of falls [46]. Its prevalence in the community ranges in some studies from 3% to others between 21% and 85% [52]. There are studies [53,54] that show that the prevalence of the fear of falling not only occurs in older adults who have suffered a previous fall, but also in older adults with no history of falls. That is why, even though most studies refer to the term “fear of falling again”, in our study the concept “fear of falling” is used, encompassing all people regardless of having suffered a fall.

In addition, in our study women suffer more falls than men; of the 245 women, 134 have fallen and of the 169 men, 54 have fallen as in other studies [46,55]. It has been estimated that women have a 58 % more likely to sustain an injury from a non-fatal fall than men [56]. Additionally, in our study, shorter and older people suffered more falls. Regarding height, significant differences have only been found between men and women [48], but not in relation to falls. Regarding age, we can say that there is a relationship between the occurrence of falls and age [57]. There are studies [58] that coincide with our results, since they affirm that people who are 80 years old suffer more falls, since they have more associated pathologies [59]. Speaking of pathologies, our results indicate that there is a relationship between having had a fall and disorders related to the nervous system, chronic psychological disorders, and disorders related to both the female and male genital tract. Within chronic disorders, one of the most common pathologies is dementia, which is related to falls, as has been found in different studies [60,61]. Urinary incontinence is one of the major geriatric syndromes with a high prevalence and a negative impact on quality of life and loss of autonomy [62]. In both men and women, the rush to get to the bathroom caused by the urgency of the moment contributes to an increased risk of falls. We have been able to verify how having had falls and the consumption of medication for the nervous system are related (163 people that had been consuming medication for the nervous system also had fallen). In the same way, a study has confirmed how older women who live in the community and receive an active medication with activity on the central nervous system, including those who take benzodiazepines, antidepressants, and anticonvulsants, present a greater risk of suffering frequent falls [62].

One of the main psychological consequences of falls is the fear of falling. Since the elderly person suffers a fall, the fear of falling again is associated with a decrease in quality of life and an increase in frailty [62]. We can verify how there is a relationship between the intensity of fear of falling and the occurrence of falls (χ^2^ = 114.74, *p* < 0.001). Our results coincide with other studies [62].

## 5. Conclusions

With our study, we have been able to understand more about the socio-demographic profile of the elderly in our community, and to analyze the risk factors related to falls which affect them. In this way, we can provide more knowledge that allows us to promote future implementation of specific, effective risk prevention and management programs, focused on this most vulnerable population group, thus promoting a multidisciplinary and effective approach in reducing falls. In this way, we can also contribute to the achievement of an active ageing model.

## Figures and Tables

**Figure 1 ijerph-20-04921-f001:**
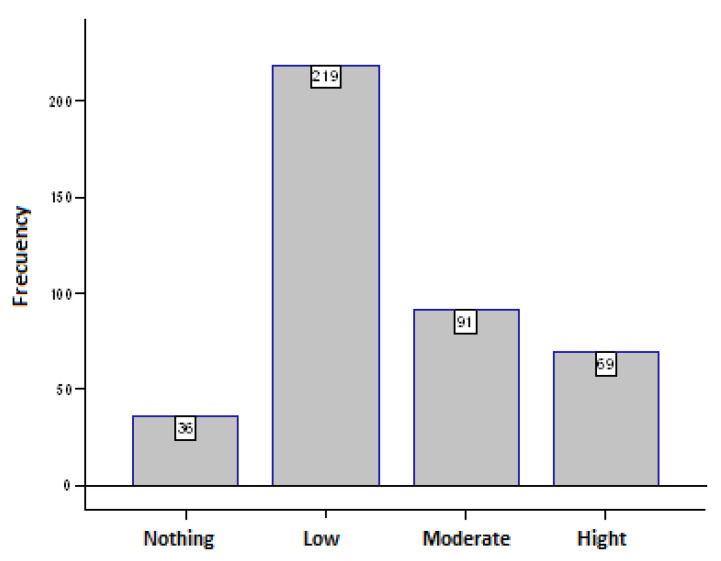
Fear of falling intensity.

**Figure 2 ijerph-20-04921-f002:**
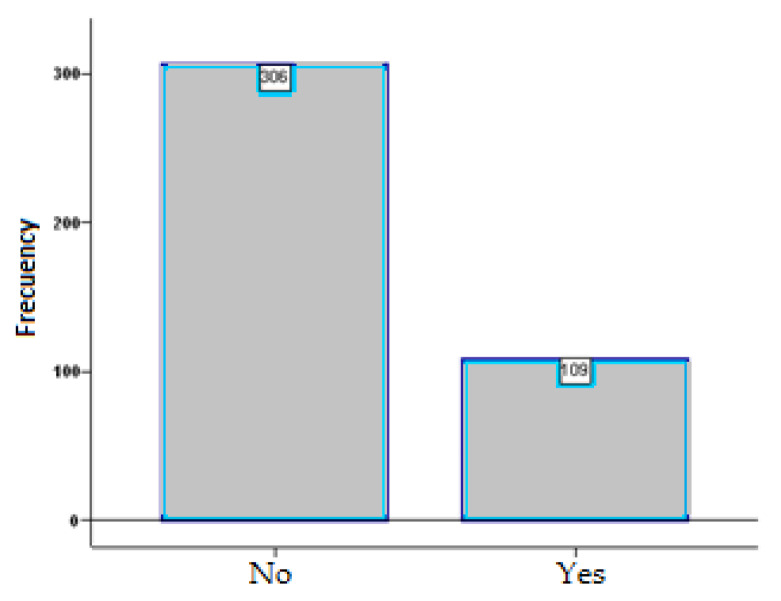
Does the fear of falling limit your activities?

**Figure 3 ijerph-20-04921-f003:**
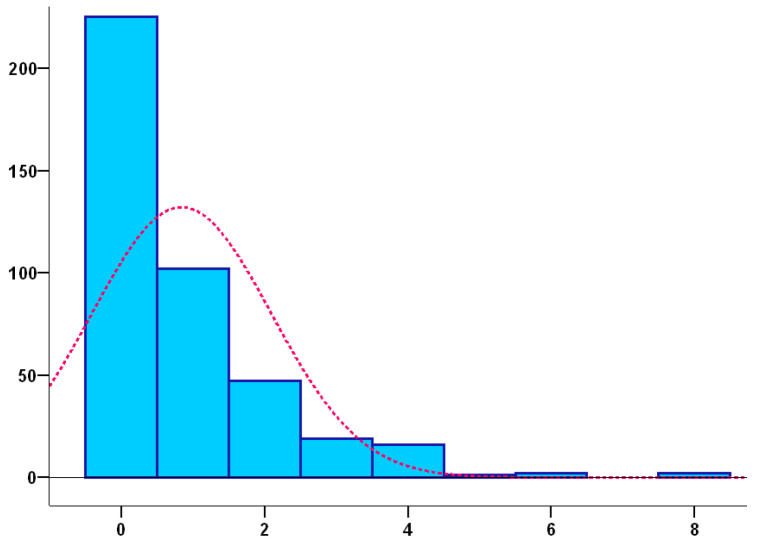
Number of falls.

**Table 1 ijerph-20-04921-t001:** Exclusion and inclusion criteria.

Exclusion Criteria	Inclusion Criteria
Being bedridden due to illness in the last three months	Both genders
Being terminally ill	Ages equal to or greater than 75 years
Have a diagnosis of severe cognitive impairment	Residents in the metropolitan district of Salamanca
Being institutionalized	Attached to the Public Health System
Being displaced within less than three months in the Basic Health Zone.	Agreed to participate
Failure to sign the informed consent to participate in the study	

**Table 2 ijerph-20-04921-t002:** Socio-demographic variables, and the age, height, and ibm descriptive values.

SOCIO- DEMOGRAPHICS		f	%
GENDER	MALE	170	41.0
	FEMALE	245	59.0
MARITAL STATUS	SINGLE	24	5.7
	MARRIED	262	63.1
	WIDOWED	129	31.1
PROFESSION	WORKER	310	74.7
	HOUSEKEEPER	105	25.3
RESIDENCE	URBAN	409	98.6
	RURAL	6	1.4
DWELLING	FLAT	407	98.1
	HOUSE	8	1.9
SUPPORT	NO	207	49.9
	YES	208	50.1
**AGE, HEIGHT, BMI**		**X**	**SD**
	AGE	82.63	5.28
	HEIGHT	1.57	0.09
	BMI	27.82	4.49

NOTE: f = Frequency. % = Percentage. Risk α = 0.05. X = Half statistic. SD = Standard Deviation.

**Table 3 ijerph-20-04921-t003:** The chronic diseases and the medication.

CHRONIC DISEASES		f	%
DIGESTIVE SYSTEM	YES	312	75.2
	NO	103	24.8
EYES AND ATTACHMENTS	YES	270	65.1
	NO	145	34.9
CIRCULATORY SYSTEM	YES	391	94.2
	NO	24	5.8
MSCULOSKELETAL SYSTEM	YES	389	93.7
	NO	26	6.3
PSYCHOLOGICAL DISORDERS	YES	208	501
	NO	207	49.9
RESPIRATORY SYSTEM	YES	327	78.8
	NO	88	21.2
EPITHELIAL APPARATUS	YES	312	75.2
	NO	103	24.8
ENDOCRINE SYSTEM, METABOLISM	YES	358	86.3
	NO	57	13.7
URINARY TRACT	YES	253	61.0
	NO	162	39.0
**MEDICATION**		**f**	**%**
MEDICATION FOR THE ALIMENTARY TRACT	YES	311	74.9
	NO	104	25.1
MEDICATION FOR THE CV SYSTEM	YES	364	87.7
	NO	51	12.3
MEDICATION FOR THE NERVOUS SYSTEM	YES	343	82.7
	NO	72	17.3

NOTE: f = Frequency. % = Percentage. Risk α = 0.05.

**Table 4 ijerph-20-04921-t004:** Existence of falls based on socio-demographic variables.

Variables	Have Had Falls		X^2^	*p*
	No	Yes		
**Gender**				
Male	115	54	20.86	<0.001
Female	111	134		
**Support**				
No support	128	78	9.42	0.002
Yes support	98	110		

**Table 5 ijerph-20-04921-t005:** Existence of falls based on socio-demographic quantitative variables.

Variables	M	SD	Est.	*p*
**Height**				
No falls	1.58	0.094	2.07 ^1^	0.038
Yes falls	1.56	0.083		
**Age**				
No falls	81.96	5.40	17277.50 ^2^	0.001
Yes falls	83.46	5.02		

Note: M = Mean, SD = Standard Deviation, Est = statistical test, *p* = statistical signification. ^1^—t Student Test ^2^—U Mann–Whitney Test. Risk α = 0.05.

**Table 6 ijerph-20-04921-t006:** Relationship between having had falls and the presence of chronic disorders.

Variables	Have Had Falls		X^2^	*p*
	No	Yes		
**Chronic Disease: Nervous System**				
No	142	84	7.46	0.006
Yes	93	95		
**Chronic disease: psychological**				
No	123	103	4.33	0.037
Yes	83	105		
**Female genital tract and breasts**				
No	202	24	11.35	0.001
Yes	145	43		
**Male genital tract and breasts**				
No	155	71	10.48	0.001
Yes	155	33		

Note: X^2^ = Chi-cuadrado de Pearson Test, *p* = Significance. Risk α = 0.05.

**Table 7 ijerph-20-04921-t007:** Relationship between having had falls and the type of medication consumed.

Variables	Have Had Falls		X^2^	*p*
	No	Yes		
**Medication for Nervous System**				
No	46	180	3.60	NS
Yes	25	163		
**Medication for Respiratory System**				
No	188	38	7.09	0.008
Yes	136	52		

Note: X^2^ = Chi-cuadrado de Pearson Tes, *p* = Significance. Risk α = 0.05.

**Table 8 ijerph-20-04921-t008:** Relationship between having had falls and the intensity of the fear of falling.

Variables	Have Had Falls		X^2^	*p*
	No	Yes		
**Intesity Fear of Falling**				
Nothing	33	3	197.63	<0.001
Low	174	44		
Moderate	17	74		
High	2	67		

*NOTE*: X^2^ = Chi-cuadrado de Pearson Test, *p* = Significance. Risk α = 0.05.

**Table 9 ijerph-20-04921-t009:** Have had falls vs. Fear of falling test, Tinetti test, Barthel Test, and Downton Test.

Variables	Have Had Falls		X^2^	*p*
	No	Yes		
**Fear of Falling Test**				
No	142	84	114.74	<0.001
Yes	21	167		
**Tinetti Test**				
Moderate risk (19–24 points)	114	61	13.32	
Severe risk (until 19 points)	113	127		<0.001
**Barthel Test**				
Independent (100 points)	32	19		
Light dependent (≥60 points)	189	154		
Moderate dependent (40–55 points)	5	13	9.03	0.029
Severe dependent (20–35 points)	0	2		
**Downton Test**				
No risk of falls (<3 points)	41	185		
With risk of falls (≥3 points)	6	182	22.79	<0.001

Note: X^2^ = Chi-cuadrado de Pearson test, *p* = Significance. Risk α = 0.05.

## Data Availability

The data presented in this study are available on reasonable request from the corresponding author. The data are not publicly available due to the applicable data protection law.

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
