# Peer review of "Living Conditions and the Incidence and Risk of Falls in Community-Dwelling Older Adults: A Multifactorial Study"

_ijerph, 2023, doi:10.3390/ijerph20064921_

Round 1

Reviewer 1 Report

ABSTRACT

- You must present the main statistical analyzes applied

- You must present, with numbers, the main results of the study

INTRODUCTION

- The first six paragraphs are confusing and unrelated to each other. I suggest a reformulation that focuses on the question of the epidemiology of aging and then links it with the issue of falls.

- If the study design is cross-sectional, talking about the incidence of falls is problematic. At most, talking about prevalence is more appropriate.

- This format of presenting specific objectives is unusual. I suggest removing or organizing them synthetically in running text.

METHODS

- Why only older people over 75?

- Who (how) assessed the presence of severe cognitive impairment? And were other levels of cognitive dysfunction, which can compromise test performance, not taken into account?

- Based on what was it established that in the study chronic diseases and consumed medications are dependent variables?

- What is the reference for the fear of falling variables?

- Describe the list with the 17 chronic diseases evaluated. Was the presence of chronic disease self-reported or was a medical diagnosis required?

- Why the last four falls? And if the subject didn't fall four times, what happened?

- Insert the references of all the instruments listed in item 2.9

- Are the values presented in table 2 already results of the study? If yes, why aren't they in the results?

- If the study is observational, what does topic 2.10 of “DESCRIPTION OF THE INTERVENTION” mean? Was there any intervention with the participants? If not, I recommend replacing the name intervention with a more appropriate one.

- A study of this nature, with a reasonable size n, no multivariate analysis was performed, with at least one of the outcomes?

RESULTS

- What does the acronym (DE) stand for?

- Have diabetes and hypertension been evaluated? It's strange they don't appear on the list of most prevalent diseases. Even because, when we look at the list of the most common medications, they are not related to the most common diseases.

- What is the difference between the result in table 2 and the result in line 280-281?

- This phrase is inappropriate “REST OF THE TESTS”. By the way, the results described in that thread are excessive and say practically nothing. This has to do with what was discussed in the previous topic, about the lack of a multivariate analysis to give robustness to the results. For example, what is the logic in regrouping variables, as in Barthel, in 5 and 3 levels?

DISCUSSION

- What does this phrase mean (According to the inclusion criteria of our study, the elderly people who participated in it stated that they were a sample according to them”?

- Where is this result (The presence of the pathology of Arterial Hypertension (HTA))?

- Where is it in the result, something that indicates this (hypnotics and anxiolytics and antidepressants have been independently associated, by 50% each, with the probability of falls)? And those too (The results obtained in our study coincide with the rest of the investigations, the most used have been those related to the cardiovascular system, followed by those that treat pathologies derived from the nervous system and finally those related to the alimentary tract and metabolism).

- the rest of the discussion follows the same pattern as the previous topic. These are generic discussions that are often not supported by the results presented.

- This topic “FUTURE LINES OF RESEARCH” makes no sense.

Author Response

RESPONSE TO REVIEWERS:

REVIEWER 1

First of all, we would like to thank you for your time in proofreading the manuscript. Your suggestions and comments have improved it considerably. We have made all the changes as you suggested. Here are the details.

ABSTRACT

- You must present the main statistical analyzes applied

Thank you very much for the input. We will make the changes as you have told us.

- You must present, with numbers, the main results of the study

Thank you very much for the suggestion. We will present the main results of the study with numbers as you have told us.

 INTRODUCTION

- The first six paragraphs are confusing and unrelated to each other. I suggest a reformulation that focuses on the question of the epidemiology of aging and then links it with the issue of falls.

Thank you very much for the inputs. We will make the changes as you have told us.

- If the study design is cross-sectional, talking about the incidence of falls is problematic. At most, talking about prevalence is more appropriate.

Thank you very much for the inputs. We will talk about prevalence instead of incidence as you have told us.

- This format of presenting specific objectives is unusual. I suggest removing or organizing them synthetically in running text.

Thank you very much for the inputs. We will remove the specific objectives as you have told us.

 METHODS

- Why only older people over 75?

Thank you very much for the suggestion.  We follow a classification of falling risk that considers from 75 years there’s a risk so for us it was important to meet this criteria.

- Who (how) assessed the presence of severe cognitive impairment? And were other levels of cognitive dysfunction, which can compromise test performance, not taken into account?

Thank you very much for the suggestion.  The presence of cognitive impairment was diagnosed by a doctor and the diagnosis was recorded in the person’s medical history. The other levels of cognitive dysfunction were taken into account but we didn’t restrict the people performing the test, if they could perform it.

- Based on what was it established that in the study chronic diseases and consumed medications are dependent variables?

Thank you very much for the suggestion. Based on the definition of dependent variable as it may be susceptible to change.

- What is the reference for the fear of falling variables?

Thank you very much for the suggestion. We will include the reference of the variables of Fear of falling.

- Describe the list with the 17 chronic diseases evaluated. Was the presence of chronic disease self-reported or was a medical diagnosis required?

Thank you very much for the suggestion.  The list of the seventeen chronic pathologies evaluated is the next, following the International Classification of Primary Care (ICPC-2):

  • Chronic disorders: general or non-specific problems.
  • Chronic disorders: blood, hematopoietic organs, immune system.
  • Chronic disorders: digestive system.
  • Chronic disorders: eyes and attachments.
  • Chronic disorders: hearing apparatus. Chronic disorders: circulatory system.
  • Chronic disorders: musculoskeletal system.
  • Chronic disorders: nervous system
  • Chronic disorders: psychological disorders
  • Chronic disorders: respiratory system Chronic disorders: epithelial apparatus
  • Chronic disorders: endocrine system, metabolism and nutrition
  • Chronic disorders: urinary tract Chronic disorders: family planning, pregnancy, childbirth, puerperium
  • Chronic disorders: genital, female and breast
  • Chronic disorders: genital, male and breast
  • Chronic disorders: social aspects

The presence of a chronic disease was known through a medical diagnosis.

- Why the last four falls? And if the subject didn't fall four times, what happened?

Thank you very much for the suggestion. It was decided in the last four falls to set a limit. The aim was to collect data on the different characteristics of falls and their consequences. If the subject did not fall four times, the lower number of falls he would have suffered or if he had not suffered any would also be noted.

- Insert the references of all the instruments listed in item 2.9

Thank you very much for the suggestion. We will insert the references of all the instruments listed in item 2.9 as you have told us.

- Are the values presented in table 2 already results of the study? If yes, why aren't they in the results?

Thank you very much for the suggestion. We will present table 2 in the results section as you have told us.

- If the study is observational, what does topic 2.10 of “DESCRIPTION OF THE INTERVENTION” mean? Was there any intervention with the participants? If not, I recommend replacing the name intervention with a more appropriate one.

Thank you very much for the suggestion. We will replace the name intervention as you have told us, we will change the tittle to “Information collection process”

- A study of this nature, with a reasonable size n, no multivariate analysis was performed, with at least one of the outcomes?

Thank you very much for the suggestion. We agree that a multivariate study may be a possibility for analysis, but we consider that the analysis we have carried out is sufficient to draw conclusions in the study.

 RESULTS

- What does the acronym (DE) stand for?

Thank you very much for the suggestion.  It was a mistake in the transcription; it means standard deviation (SD).

- Have diabetes and hypertension been evaluated? It's strange they don't appear on the list of most prevalent diseases. Even because, when we look at the list of the most common medications, they are not related to the most common diseases.

Thank you very much for the suggestion.  Yes, both diseases have been evaluated. In the case of diabetic pathologies: 358 (86.3%) have suffered from the disease and 57 (13.7%) have not. In the case of diseases related to the circulatory system, such as hypertension: 391 (94.2%) have suffered from the disease and 24 (5.8%) have not.

- What is the difference between the result in table 2 and the result in line 280-281?

Thank you very much for the suggestion.  Table 2 represents the descriptive values of the psychometric tests and the line 280-281 refers the chronic diseases that the people have in the study.

- This phrase is inappropriate “REST OF THE TESTS”. By the way, the results described in that thread are excessive and say practically nothing. This has to do with what was discussed in the previous topic, about the lack of a multivariate analysis to give robustness to the results. For example, what is the logic in regrouping variables, as in Barthel, in 5 and 3 levels?

Thank you very much for the suggestion.  We will change the phrase as you have told us and we will be more practically with that paragraph.

About reagrouping variables, we did it like this way because some of the sections of the classification of each test were a minority represented by the participants of our study, as is the case of severe risk in the case of the Barthel test.

 DISCUSSION

- What does this phrase mean “According to the inclusion criteria of our study, the elderly people who participated in it stated that they were a sample according to them”?

Thank you very much for the suggestion.  According to the inclusion and exclusion criteria we have developed, the participants in the study appear to be a sample according to the inclusion criteria.

- Where is this result “The presence of the pathology of Arterial Hypertension (HTA)”?

Thank you very much for the suggestion.  We will add the result of the presence of the pathology of Arterial Hypertension, 391 people (94.2%) have had the disease.

- Where is it in the result, something that indicates this (hypnotics and anxiolytics and antidepressants have been independently associated, by 50% each, with the probability of falls)? And those too (The results obtained in our study coincide with the rest of the investigations, the most used have been those related to the cardiovascular system, followed by those that treat pathologies derived from the nervous system and finally those related to the alimentary tract and metabolism).

Thank you very much for the suggestion. We will indicate in the results the data as you have told us.

- The rest of the discussion follows the same pattern as the previous topic. These are generic discussions that are often not supported by the results presented.

Thank you very much for the suggestion. We will support the results as you have told us.

- This topic “FUTURE LINES OF RESEARCH” makes no sense.

Thank you very much for the suggestion. We will retire this point.

Reviewer 2 Report

This article is effective in preventive medicine, especially in the older adults, falls are

the main accidental injuries at present, and the entire article still has the following

missing points: (Please revise the full text substantially)

1. Please use male and female for gender.

2. The content will attract readers, but the whole article lacks structure and seems to

have too many paragraphs. For example, the independent variable and dependent

variable can be displayed in Table.

3. The abstract is described in one paragraph.

4. Please present the discussion in the form of an article. At present, it looks like a

question and answer.

5. Many cited documents are too old, whether there are documents in the past five

years.

6. Please use MDPI format.

Author Response

REVIEWER 2

This article is effective in preventive medicine, especially in the older adults, falls are the main accidental injuries at present, and the entire article still has the following missing points: (Please revise the full text substantially)

First of all, we would like to thank you for your time in proofreading the manuscript. We have made all the changes as you suggested, which has made the manuscript much more correct. Here are the details.

  1. Please use male and female for gender.

Thank you very much for the input. We will use it as you have told us.

  1. The content will attract readers, but the whole article lacks structure and seems to have too many paragraphs. For example, the independent variable and dependent variable can be displayed in Table.

Thank you very much for the suggestion. We will make the changes as you have told us.

  1. The abstract is described in one paragraph.

Thank you very much for the input. We will make the changes as you have told us.

  1. Please present the discussion in the form of an article. At present, it looks like a question and answer.

Thank you very much for the suggestion. We will present it as you have told us.

  1. Many cited documents are too old, whether there are documents in the past five years.

Thank you very much for the input. We will make the changes as you have told us.

  1. Please use MDPI format.

Thank you very much for the suggestion.  We will use the MDPI format.

Reviewer 3 Report

This article explores the relationship between the incidence and risk factors of falls in elderly adults living in the community. Overall, the article tackles an interesting and important topic, and the number of involved participants is good, but the language and writing style makes the whole study difficult to follow. Furthermore, the message of this article is completely unclear. The authors present too many results, and it is difficult to understand which ones are important. Some suggestions for improvement are provided below.

Comments/suggestions for improvement:

Title

The title is too long and written in a confusing way. I suggest rewriting it to something like this: "Living conditions and the incidence and risk of falls in community-dwelling older adults: A multifactorial study". This title revision would make the article's content much clearer for the readers.

Abstract

The abstract is too long, it should have a maximum of 200 words, and there should be no headings in the abstract (such as “background”, “methods”, etc.). Furthermore, there should be no references to other sources (citations) in the abstract. The abstract should be concise and not explain each step in the research process – e.g., the sentences from line 23 to line 26 (“Then, the field work is conducted…” and “Once the results obtained…”) is completely redundant. The message of the abstract is very unclear; what are the main results exactly? And what is the conclusion of this study? The meaning of the last lines of the abstract (33 and 34) is unclear; there is a writing mistake.

Introduction

Many sentences are too long and difficult to follow. For example, the first sentence in the introduction could be split into two sentences, maybe even three. It is also unclear why sentences are sometimes split into different paragraphs when they belong to the same theme. There is also a lot of unnecessary general information. The introduction should be more concise – introduce the problem and situate your approach within the wider body of research.

Furthermore, numerous grammatical and writing errors make certain sentences difficult to read. The paragraphs from line 72 to line 83 could all be placed into one paragraph and shortened. Also, the sentence starting at line 85 (Falls in the elderly…) is too long. It should be split into at least two sentences, if not more.

It is not necessary to state your hypothesis in the introduction. The presented hypothesis is very complicated and with too many variables. The research objectives can stay.

Materials and Methods

It is unclear why this section was written in bullet points and not in classic textual form. If many points need to be presented, I suggest using a table format rather than having so many subtitles and bullet points. It is difficult to read and very repetitive. It would be helpful to clarify why the 9-point Likert scale was selected. Also, references should be given for the listed evaluation tools (Fear of Falling Test, Tinetti Test, etc.). It is unclear why each sentence in section 2.10 is a separate paragraph. They could all be in the same paragraph; the text could be shortened.

Results

This is the section that is the most difficult to follow, and it should be the most important part of the article. There are so many pages of subtitles and minor results presented under each subtitle. The problem with this is that it becomes unclear which results are important and where the reader should focus their attention. I suggest a completely different approach – to present the majority of your results via tables and graphs. The text can accompany them and describe some major findings, but not every single result should be presented in this paper. It seems that the authors are trying to put too many results into this publication; the most important results are not easy to identify.

Discussion

The Discussion has the same problem as the results. As too many different results are presented, the Discussion attempts to discuss all of them under different subtitles. The Discussion should be more coherent with something like 3-4 main discussed themes with their subtitles. The problem stems from the Results section, the authors need to identify and highlight only the important results, not every single minor result. They should be grouped into larger themes (3-4), that can then be discussed further in the Discussion section. As a reader, I am not sure what this article's main message is.

Conclusion

The conclusion should not repeat/summarize the results again but discuss the implications of the results for different stakeholders in the care field and provide an outlook for further research.

Thank you for the opportunity to review your work. I hope that my comments are helpful. 

Author Response

REVIEWER 3

This article explores the relationship between the incidence and risk factors of falls in elderly adults living in the community. Overall, the article tackles an interesting and important topic, and the number of involved participants is good, but the language and writing style makes the whole study difficult to follow. Furthermore, the message of this article is completely unclear. The authors present too many results, and it is difficult to understand which ones are important. Some suggestions for improvement are provided below.

First of all, we would like to thank you for your time in proofreading the manuscript. We would also like to thank you for your comments, which have been of great help in improving the manuscript. We have made all the changes as you suggested. Here are the details.

 Comments/suggestions for improvement:

 Title

The title is too long and written in a confusing way. I suggest rewriting it to something like this: "Living conditions and the incidence and risk of falls in community-dwelling older adults: A multifactorial study". This title revision would make the article's content much clearer for the readers.

Thank you very much for the suggestion.  We will change the title as you have told us.

Abstract

The abstract is too long, it should have a maximum of 200 words, and there should be no headings in the abstract (such as “background”, “methods”, etc.). Furthermore, there should be no references to other sources (citations) in the abstract. The abstract should be concise and not explain each step in the research process – e.g., the sentences from line 23 to line 26 (“Then, the field work is conducted…” and “Once the results obtained…”) is completely redundant. The message of the abstract is very unclear; what are the main results exactly? And what is the conclusion of this study? The meaning of the last lines of the abstract (33 and 34) is unclear; there is a writing mistake.

Thank you very much for the input. We will make the changes as you have told us.

 Introduction

Many sentences are too long and difficult to follow. For example, the first sentence in the introduction could be split into two sentences, maybe even three. It is also unclear why sentences are sometimes split into different paragraphs when they belong to the same theme. There is also a lot of unnecessary general information. The introduction should be more concise – introduce the problem and situate your approach within the wider body of research.

Furthermore, numerous grammatical and writing errors make certain sentences difficult to read. The paragraphs from line 72 to line 83 could all be placed into one paragraph and shortened. Also, the sentence starting at line 85 (Falls in the elderly…) is too long. It should be split into at least two sentences, if not more.

It is not necessary to state your hypothesis in the introduction. The presented hypothesis is very complicated and with too many variables. The research objectives can stay.

Thank you very much for the suggestion. We will be more concise and we will write the paragraphs shortened and also retire the hypothesis of the introduction.

 Materials and Methods

It is unclear why this section was written in bullet points and not in classic textual form. If many points need to be presented, I suggest using a table format rather than having so many subtitles and bullet points. It is difficult to read and very repetitive. It would be helpful to clarify why the 9-point Likert scale was selected. Also, references should be given for the listed evaluation tools (Fear of Falling Test, Tinetti Test, etc.). It is unclear why each sentence in section 2.10 is a separate paragraph. They could all be in the same paragraph; the text could be shortened.

Thank you very much for the suggestion.  We will use a table formant, clarify the scale, add the references for the listed evaluation tools and put together the paragraphs in section 2.10 besides write it shortened, as you have told us.

 Results

This is the section that is the most difficult to follow, and it should be the most important part of the article. There are so m

any pages of subtitles and minor results presented under each subtitle. The problem with this is that it becomes unclear which results are important and where the reader should focus their attention. I suggest a completely different approach – to present the majority of your results via tables and graphs. The text can accompany them and describe some major findings, but not every single result should be presented in this paper. It seems that the authors are trying to put too many results into this publication; the most important results are not easy to identify.

Thank you very much for the suggestion. We will present the results via tables and graphs and without every single result, as you have told us.

 Discussion

The Discussion has the same problem as the results. As too many different results are presented, the Discussion attempts to discuss all of them under different subtitles. The Discussion should be more coherent with something like 3-4 main discussed themes with their subtitles. The problem stems from the Results section, the authors need to identify and highlight only the important results, not every single minor result. They should be grouped into larger themes (3-4), that can then be discussed further in the Discussion section. As a reader, I am not sure what this article's main message is.

Thank you very much for the input. We will make the changes as you have told us.

 Conclusion

The conclusion should not repeat/summarize the results again but discuss the implications of the results for different stakeholders in the care field and provide an outlook for further research.

Thank you very much for the suggestion. We will make the changes as you have told us.

Round 2

Reviewer 1 Report

 - The authors complied with some of the suggested revisions. However, the article still has serious design and methodological problems, as follows:

- The explanation about the issue of age over 75 did not convince me. After all, if the authors report that being over 75 years old is a risk factor, it would be essential to compare the risk of these with younger people (between 60 and 75 years old).

- I didn't understand the answer to that question (Based on what was it established that in the study chronic diseases and consumed medications are dependent variables?)

- All the tables that were inserted, associating falls and comorbidities, could be in a single table. Furthermore, there is an exhaustive description of the results in the text which adds nothing.

- Why was there no analysis of sociodemographic variables with falls?

- Regarding the issue of multivariate analysis, I totally disagree. From an epidemiological point of view, it is fundamental that the results are submitted to a more rigorous analysis in order to identify the factors associated with the outcome in a more adequate way.

In addition, other point worth highlighting:

- The figures inserted (about FEAR OF FALLING TEST, Does the fear of falling limit your activities? and the histogram of the total number of falls) are not numbered, nor are they cited in the body of the text of the results. By the way, from the information they bring, this could just be in the text, without the need for tables.

Author Response

REVIEW 1

 The authors complied with some of the suggested revisions. However, the article still has serious design and methodological problems, as follows:

- The explanation about the issue of age over 75 did not convince me. After all, if the authors report that being over 75 years old is a risk factor, it would be essential to compare the risk of these with younger people (between 60 and 75 years old).

Thank you very much for your comments, perhaps we have expressed ourselves badly. The fact that the inclusion criteria for people over 75 years of age has to do with the increased risk associated with this age in the scientific literature. We wanted to make an exhaustive analysis of this risk. But the question you raise is very correct, in future studies it would be very interesting to compare this risk or the etiological factors that lead to an increase in this risk with people of other ages, perhaps between 65 and 75 years of age. Thank you very much for your contribution, we will take it into account in the near future.

- I didn't understand the answer to that question (Based on what was it established that in the study chronic diseases and consumed medications are dependent variables?)

Thank you very much for the input. The description may lead to misunderstandings. We modify the text in the method section so that it is not misleading.

- All the tables that were inserted, associating falls and comorbidities, could be in a single table. Furthermore, there is an exhaustive description of the results in the text which adds nothing.

Thank you very much for the suggestion. We will make the changes as you have told us, we will make only a table.

- Why was there no analysis of sociodemographic variables with falls?

Thank you very much for the input. Yes, there is analysis of socio-demographic variables with falls, we explain in the page eight the main results (the gender, having support, height, age).

- Regarding the issue of multivariate analysis, I totally disagree. From an epidemiological point of view, it is fundamental that the results are submitted to a more rigorous analysis in order to identify the factors associated with the outcome in a more adequate way.

Thank you very much for the suggestion. We are sure that a multivariate analysis would also provide important results, but in this case we have considered carrying out all the analyses shown in the methodological section, which are then presented in the results. Nevertheless, in the future we are considering carrying out this type of analysis to obtain scientific results. We welcome your comments.

In addition, other point worth highlighting:

- The figures inserted (about FEAR OF FALLING TEST, Does the fear of falling limit your activities? and the histogram of the total number of falls) are not numbered, nor are they cited in the body of the text of the results. By the way, from the information they bring, this could just be in the text, without the need for tables.

Thank you very much for the suggestion. We will make the changes as you have told us, we will retire the tables of this point.

Reviewer 2 Report

I suggest publishing it in its current form

Author Response

REVIEW 2

-I suggest publishing it in its current form.

Thank you very much for the suggestion and for your assessment, we really appreciate your attention.

Reviewer 3 Report

Thank you for making the requested changes, the manuscript is improved compared to the previous version. 

However, there are still multiple language mistakes and unusual sentence structures in the manuscript. I pointed out some of the issues on the first two pages of the manuscript and at some points throughout the text. There are many more issues besides the ones I highlighted. I strongly advise the authors to have the whole manuscript checked by a native speaker.

I attached a pdf document with direct comments on the manuscript file, I hope that these comments are helpful.

Author Response

REVIEW 3

-Thank you for making the requested changes, the manuscript is improved compared to the previous version. 

Thank you for appreciating the change we have made and for your attention and suggestions.

However, there are still multiple language mistakes and unusual sentence structures in the manuscript. I pointed out some of the issues on the first two pages of the manuscript and at some points throughout the text. There are many more issues besides the ones I highlighted. I strongly advise the authors to have the whole manuscript checked by a native speaker.

Thank you very much for the suggestion. We will correct the language mistakes and sentence structures that you have pointed in the manuscript and also the highlighted issues.

I attached a pdf document with direct comments on the manuscript file, I hope that these comments are helpful.

Thank you very much for the suggestion. We will make the changes as you have told us in the pdf document, it will be very helpful for us.
